# Design and Characteristic Analysis of an Axial Flux High-Temperature Superconducting Motor for Aircraft Propulsion

**DOI:** 10.3390/ma16093587

**Published:** 2023-05-07

**Authors:** Jun-Yeop Lee, Gi-Dong Nam, In-Keun Yu, Minwon Park

**Affiliations:** 1Department of Electrical Engineering, Changwon National University, Changwon 51140, Republic of Korea; 2Korea Electrotechnology Research Institute, Changwon 51543, Republic of Korea; 3Institute of Mechatronics, Changwon National University, Changwon 51140, Republic of Korea

**Keywords:** aircraft propulsion system, axial flux motor, superconducting rotating machine, 2G HTS wire, GdBCO wire

## Abstract

In line with global environmental regulations, the demand for eco-friendly and highly efficient aircraft propulsion systems is increasing. The combination of axial flux motors and superconductors could be a key technology used to address these needs. In this paper, an axial flux high temperature superconducting (HTS) motor for aircraft propulsion was designed and its characteristics were analyzed. A 2G HTS wire with high magnetic flux characteristic was used for the field winding of the 120 kW axial flux HTS motor, and the rotational speed and rated voltage of the motor were 2000 rpm and 220 V, respectively. The axial flux HTS motor implements a revolving armature type for solid cooling of the HTS field coil. The electromagnetic and thermal features of the motor were analyzed and designed utilizing a 3D finite element method program. The HTS coil was maintained at the target temperature by effectively designing the current lead and cooling system to minimize heat loss. These results can be effectively used in the design of propulsion systems for large commercial aircraft in the future as well as for the design of small aircraft with less than 4 seats.

## 1. Introduction

Recently, as regulations on exhaust gases that cause environmental pollution have been strengthened worldwide, the demand for eco-friendly and highly efficient transportation means is increasing. In the case of electric propulsion vehicles, which are in the limelight as an eco-friendly propulsion system, they can be of great help to environmental problems due to their non-polluting and low-noise characteristics. However, it is difficult to replace the existing aircraft propulsion system with electricity-based propulsion systems because of their low energy density and large and heavy motors. Therefore, various energy sources and motor technologies are being studied to reduce volume and weight [1,2]. An axial flux motor-based propulsion system is one of the effective ways to reduce the weight of the aircraft propulsion system. Axial flux motors have a structural characteristic in which the conductors are arranged radially, and the stator and rotor are disc-shaped [3]. Because of these structural features, the magnetic flux of the axial flux motor moves in the axial direction, and compared to radial type flux motors, it has the advantages of less leakage flux, higher torque-to-weight ratio, higher efficiency, and less heat generation [4,5].

In aircraft propulsion systems, the miniaturization of motors can improve propulsion efficiency by optimizing the propulsion system and fuselage structure to reduce fluid resistance. Superconducting technology is an effective way to reduce the size of motors by increasing power density. Superconducting wires can obtain higher current density at a specific temperature than copper wires, enabling the realization of compact and high power density motors [6,7,8]. Therefore, if a superconducting coil is applied to an existing axial motor, the advantages of light weight, miniaturization, and high efficiency can be maximized [9,10,11]. Synchronous motors are mostly designed with a fixed armature type due to the weight of the armature winding or the economical reason for choosing a slip ring. However, rotating the field part of an high temperature superconducting (HTS) motor can be burdensome due to the need for maintaining a vacuum and cryogenic environment. Therefore, The axial flux HTS motor presented in this paper uses a rotating armature type, which has several advantages. These include a relatively simple cooling system design, stable vacuum maintenance, and easy power supply because the HTS field part is fixed. The axial flux HTS motor uses copper coils to replace the low durability and uncertain performance equivalence of permanent magnets, and consists of a combination of HTS coils, which are easier to fabricate than HTS bulk. The cooling system of electric aircraft propulsion motors has been studied in various topologies such as spray cooling, heat pipe cooling, and cooling jacket to cool the heat caused by winding losses and core losses (eddy current losses, hysteresis losses) [12,13,14]. The cooling system of the axial flux HTS motor is divided into the HTS field and armature parts. The cooling system of the superconducting stator part is a forced convection cooling method using a cryogenic blower, which must be able to maintain the operating temperature of the HTS stator coil stably. For the armature core, which is made of GFRP, a relatively simple air cooling method is adopted because there is no core loss.

In this paper, the design and characteristic analysis of an axial flux HTS motor for aircraft propulsion are dealt with. To construct a stable and effective cooling system for the HTS field coil, a fixed field type 120 kW axial flux motor was designed and its characteristics were analyzed in detail. The rated operating temperature and rotational speed of the designed motor are 35 K and 2000 rpm, respectively. The HTS field coil consisted of 8-pole quadruple pancake coil (QPC) and was cooled by circulating helium gas through copper pipes located between each double pancake coil (DPC). GdBCO wire was used for the superconducting field coil in consideration of performance and mechanical durability, and the critical current was selected considering the maximum bending diameter, operating temperature, magnetic field, and angle dependence that affect the critical current of the wire. The cooling system was designed to minimize the thermal load applied to the stator components in order to maintain the operating temperature, which is one of the main factors determining the performance of the superconducting field coil. The electromagnetic and thermal characteristics of the designed motor were analyzed in detail through the utilization of a 3D finite element method program. As a result of the analysis, the critical current of the designed field coil was 239 A at the operating temperature, the operating current considering the cooling margin was 150 A, and the magnetic field of the air gap was 0.45 T. The output torque and power of the motor were 580 N·m and 120 kW, respectively. As a result of thermal analysis, the maximum temperature of the cooling system surface was 33 K, and the maximum temperature of the internal cooling pipe was 32.4 K, which was within the target temperature of 35 K. Also, the helium gas was stably cooled to 26 K in a heat exchanger that recools the helium gas by liquid neon. These results show that the axial flux HTS motor has thermal stability. The results of this research are expected to be valuable in the future development of environmentally friendly aircraft propulsion systems.

## 2. Design of an Axial Flux HTS Motor for Aircraft Propulsion

### 2.1. Fundamental Structure Design of Axial Flux HTS Motor

Figure 1 shows the structure of a typical axial flux synchronous motor.

The basic structure of the axial flux HTS motor was designed with reference to Figure 1 and Equations (1)–(4). In Figure 1, $D_{o}$, $D_{i}$, $L_{i}$, g and h denote the outer diameter of the rotor, the inner diameter of the rotor, the effective length in the radial direction of the rotor, the air gap length, and the number of rotors, respectively. Other important parameters in this motor are the stator average diameter ($D_{ev}$) and the pole pitch at the average diameter (τ), and calculated as follows:(1)$$D_{o}=\sqrt[{3}]{\frac{\varepsilon P_{out}}{\Pi^{2}k_{D}k_{\omega 1}n_{s}B_{mg}A_{m}\eta cos\phi }}$$
(2)$$D_{i}=k_{D}D_{o}$$
(3)$$D_{av}=0.5(D_{o}+D_{i})$$
(4)$$\tau =\frac{\pi D_{av}}{2p}$$

In the above equations, ε is represents the phase-to-phase voltage ratio of EMF, $P_{out}$ represents the shaft power, $k_{D}$ is the coefficient based on $D_{i}/D_{o}$, $k_{\omega 1}$ represents the winding factor, $n_{s}$ represents the rotor speed, $B_{mg}$ represents the peak value of magnetic flux density in the air gap, $A_{m}$ represents the peak value of the line current density, η represents the efficiency and cosϕ is the power factor. and $k_{d}$ is the ratio of inner diameter to outer diameter of stator [15].

Figure 2 shows the structure of the axial flux HTS motor designed in this study.

The stator is composed of the field coil and cryostat, and the rotor is composed of the rotor body, the armature coil, and the magnetic shield. The operating temperature of the stator section is 35 K, which is a cryogenic environment. The cooling of the stator was achieved by circulating helium gas through the cooling pipe of the cooling plate located between the DPC coils, which facilitates the maintenance of the cooling system. Since the Joule heat generated in the armature coil is naturally cooled, the structure of the rotor is simple. The HTS field coil was designed with a fan shaped QPC, while a metal insulator was employed to ensure the stability and response time of the coil in response to changes in current. The cryostat was made of stainless steel with little structural deformation in cryogenic environments.

In order to decrease weight, the rotor body and teeth were constructed from GFRP, a non-magnetic material. Meanwhile, the magnetic shield was produced with M-27 24 Ga, which serves to increase the magnetic flux density of the rotor while reducing the impact of external magnetic fields. Table 1 shows the design specifications of the axial flux HTS motor. The rated rotating speed of the rotor is 2000 rpm in the steady state, and the number of poles is 8.

### 2.2. Design of the HTS Field Coil for the Axial Flux HTS Motor

Table 2 shows the specifications of the GdBCO wire used in the axial flux HTS motor field coil.

The field coil of the 120 kW axial flux HTS motor was designed using GdBCO wire, the second-generation HTS wire manufactured by SuNam Co., Ltd. The GdBCO wire was selected in this study because it has high critical current density, low dependency of the critical current on the external magnetic field, and good mechanical durability [16,17,18,19,20].

Figure 3 shows the $I_{c}$-*B* curves of the GdBCO wire. The critical characteristics of a superconducting wire depend on the magnitude of magnetic field and the angle of incidence.

Since the GdBCO wire is a tape type that is greatly affected by the perpendicular magnetic field, the critical current was selected based on the perpendicular magnetic field.

Figure 4 shows the magnetic field distributions of the field coil of the axial flux HTS motor.

The HTS field coil consists of 8-pole QPC. In Figure 4 The total length of the HTS wire used for the field coil is 2.09 km. The maximum magnetic field point and the maximum perpendicular magnetic field point of the field coil were located at the inner turn of the coil and the upper turn of the coil, respectively. At the operating temperature of 35 K, the critical current of the superconducting coil was 239 A, and the operating current was designed to be 150 A considering the safety margin of 63% [21].

### 2.3. Design of the Cooling System for the Axial Flux HTS Motor

Figure 5 shows the configuration of the cooling system of the axial flux HTS motor.

The cooling system of the axial flux HTS motor is divided into a baffle layer, heat pipes, a cooling pipe, and a cryogenic blower. The baffle layer is used to minimize convective heat loss, and the cryogenic blower circulates helium gas at a constant flow rate.

The neon inside the tank is cooled by heat pipes connected to cryogenic coolers and accumulates in a liquid state. This liquid neon cools the helium gas, the cryogenic refrigerant of the motor. The cooled helium gas circulates through the cooling pipe to cool the HTS field of the motor.

### 2.4. Design of the Current Lead for the Axial Flux HTS Motor

The current lead, composed of copper or brass, is responsible for providing excitation current to the field coil. Due to the Joule heat generated by conductor resistance, current leads have a high heat load. As such, the cooling system design must prioritize the reduction of heat load on stator components. Achieving this requires careful selection of current lead length and cross-sectional area to minimize heat load. Equations (5) and (6) can be used to calculate the minimum heat load and optimal shape of the current lead.
(5)$$Q_{L}=I\sqrt{2\int_{T_{L}}^{T_{H}}\rho \left(T\right)k\left(T\right)dT}$$
(6)$$\frac{L}{A}=\frac{1}{I}\int_{T_{L}}^{T_{H}}\frac{k\left(T\right)}{\sqrt{2\int_{T_{L}}^{T_{H}}\rho \left(T\right)k\left(T\right)dT}}dT$$
where, $Q_{L}$ represents the least amount of heat generated at the cold end, $T_{H}$ and $T_{L}$ are the warm-end temperature and cold-end temperature, ρ(T) is the electrical resistivity and k(T) is the thermal conductivity, I is the operating current, A is the cross section of current lead and L is the current lead length [22]. 

The specifications of the designed current lead are presented in Table 3.

The current leads are consisted of a normal conductive materials, so copper or brass is used. The current leads include current terminals, current feedthroughs, and a copper block.

## 3. Analysis Results and Discussions

### 3.1. Electromagnetic Analysis of the Axial Flux HTS Motor

Figure 6 shows the magnetic field and perpendicular magnetic field distribution for the 1/2 model of the axial flux HTS motor. 

The maximum magnetic field of the axial flux HTS motor was 1.25 T. The magnitude of the air-gap magnetic field affecting the output power of the axial flux HTS motor is 0.45 T. 

The critical current density applied to the HTS field coil made of tape-type wire varies depending on the magnetic flux density in the perpendicular direction of the winding surface present in the straight and curved portions of the field coil. Therefore, the operating current of the HTS field coil is determined by the value of the perpendicular magnetic field, which is directly related to the performance of the axial flux HTS motor.

The maximum perpendicular magnetic field generated by the axial flux motor is 1.16 T, which shows that the axial flux HTS motor can perform stably at the operating current of 150 A of the HTS field coil considering the safety margin of 63%.

Figure 7 shows the output torque of the axial flux HTS motor and the magnetic field distribution in the air gap. 

In Figure 7a, the output torque of the motor converges at 580 Nm, which indicates that the motor can produce a stable output. In Figure 7b, at a distance of 12.5 mm between the armature and the field, the magnetic field was about 0.45 T.

### 3.2. Thermal Analysis of the Axial Flux HTS Motor

Figure 8 shows the temperature distribution of the cooling system (a) and cooling pipe (b) of the axial flux HTS motor when the operating current of 150 A is applied to the current lead. 

The maximum temperature in the cooling system was 33 K at the ends of the current leads. The maximum temperature of the surface of the cooling pipe inserted into the cooling system was 32.4 K at the temperature rising part by the current lead, and the helium gas temperature after circulating the cooling pipe was 29.4 K. These results indicate that the axial flux HTS motor stably performs within the target operating temperature of 35 K for all maximum temperatures in the cooling system. The radiant heat load generated by the cooling system is 7.8 W. The radiant heat load can be reduced by increasing the multilayer insulation of the cooling system.

Figure 9 shows the temperature distribution of the heat exchanger. 

The maximum temperature of the helium gas circulating along the cooling pipe was 30 K. As a result of the simulation, it can be seen that the 30 K helium gas was cooled to 26 K again in the heat exchanger in the liquid neon chamber, and stable heat exchange was achieved.

### 3.3. Performance Test of the Axial Flux HTS Motor

Figure 10 shows the no-load characteristic curve and the short-circuit characteristic curve of the axial flux HTS motor. 

The basic characteristics of the axial flux HTS motor were verified by running open- and short-circuit simulations using the FEM program model at its rated rotating speed of 2000 rpm. The open-circuit test is conducted with the load-side circuit open and the rotor operating at rated speed to find out the relationship between field current and voltage under no-load conditions. The armature terminal voltage reached 220 V at the rated field coil current of 150 A under the motor’s rated rotating speed of 2000 rpm. The short-circuit test is a test to find out the relationship (synchronous impedance) of short-circuit current to field current, and assumes a situation in which the load-side terminal is short-circuited and the rotor is operated at the rated speed of 2000 rpm. The field coil current was gradually increased until the armature current reached to the motor’s rated current. When the field current reached 154 A, the armature current increased proportionally and eventually reached its rated value of 107.5 Arms.

Through the no-load characteristic curve and the short-circuit characteristic curve, the synchronous impedance and short-circuit ratio, which can infer the overall characteristics of the motor, such as voltage drop, stability, and efficiency, can be known. Synchronous impedance is the ratio of the motor’s terminal voltage to the armature current, and its value fluctuates according to the degree of saturation of the iron core, so it is not constant. The short-circuit ratio is the ratio of the excitation current that generates the rated voltage during the open-circuit test and the excitation current that generates the rated current during the short-circuit test. The short-circuit ratio of the motor calculated at this time was 0.974.

## 4. Conclusions

This paper dealt with the design and characteristics analysis of an axial flux HTS motor for aircraft propulsion. The design process for the axial flux HTS motor using GdBCO wire involved referencing [23] to establish the motor’s intended purpose and output, followed by taking into account the interdependent relationships between each parameter to determine the motor’s size, coil specifications, voltage, and other relevant variables. This motor was designed as a revolving armature type for stable cooling and easy maintenance of the HTS field coil. The diameter and the axial length of the axial flux HTS motor and the length of the HTS wire required were 0.6 m, 0.4 m, and 2.08 km, respectively. When designing the superconducting field coil, factors such as the maximum radius of curvature of GdBCO wire, temperature, magnetic field, and magnetic field incident angle were taken into account as these parameters can affect the critical current. The current lead shape was determined considering the minimum thermal load according to temperature conditions and operating current. As a result, the power density of the axial flux HTS motor was 1.52 kW/kg and the continuous output power was 120 kW. The maximum magnetic field and vertical magnetic field of the HTS motor were 1.25 T and 1.16 T, respectively. The maximum temperature of the stator when operating the axial flux HTS motor was 33 K, which is lower than the target operating temperature. The total radiated heat load on the stator was 7.6 W. Open and short-circuit simulations were performed to obtain the no-load and short-circuit characteristic curves, which represent the basic characteristics of the designed motor. The short-circuit ratio of this motor was 0.974. 

Axial flux structures are characterized by higher power density compared to radial flux structures [4,5], a feature also observed in aircraft propulsion motors [24]. While HTS motors can generally achieve high power density [25,26], this is not always the case for small capacities. Due to these characteristics of axial flux structures and HTS motors, the axial flux HTS motor presented in this paper can have sufficient competitiveness in terms of power density when the capacity is increased. Based on these results, it is expected that the axial flux HTS motor can be effectively used not only in the propulsion system of a small aircraft with four or fewer seats or unmanned aircraft, but also in a large commercial aircraft.

## Figures and Tables

**Figure 1 materials-16-03587-f001:**
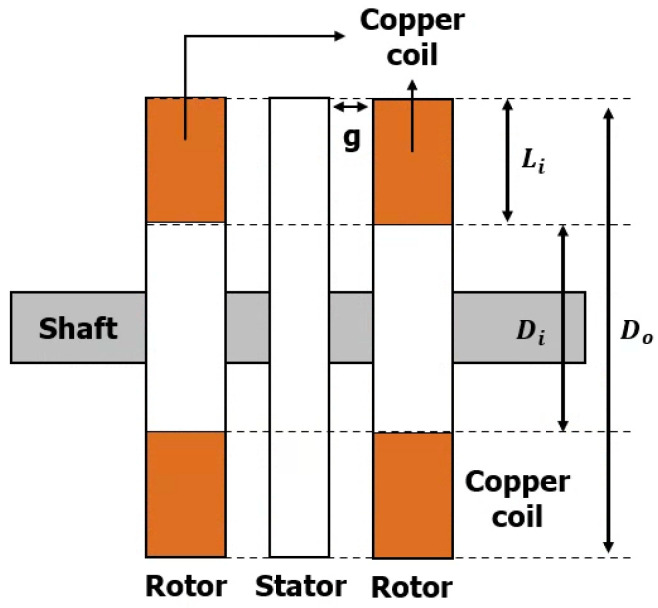
Generalized representation for an axial flux synchronous motor.

**Figure 2 materials-16-03587-f002:**
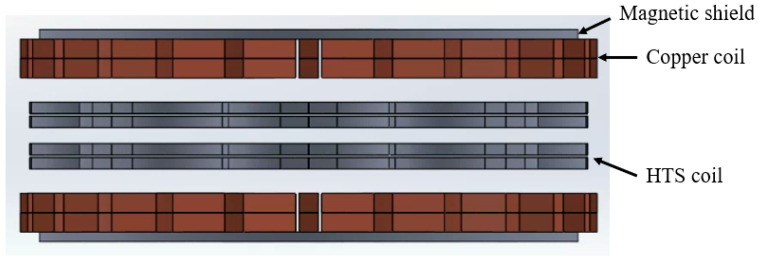
The structure of the axial flux HTS motor.

**Figure 3 materials-16-03587-f003:**
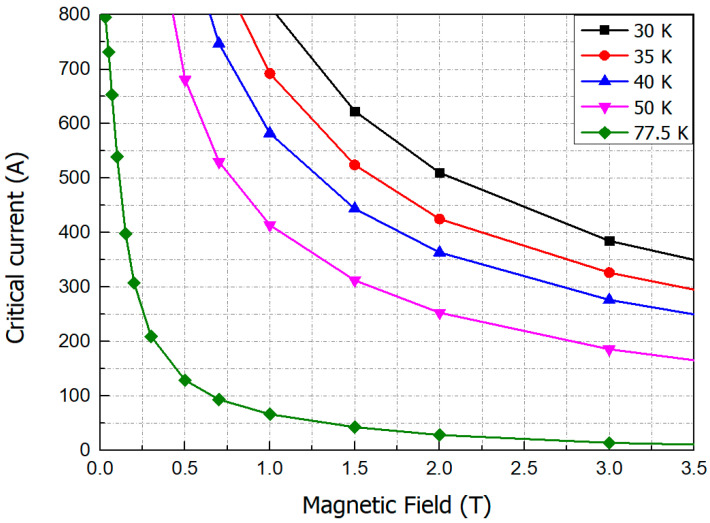
*Ic-B* curves of the GdBCO wire depending on the operating temperature.

**Figure 4 materials-16-03587-f004:**
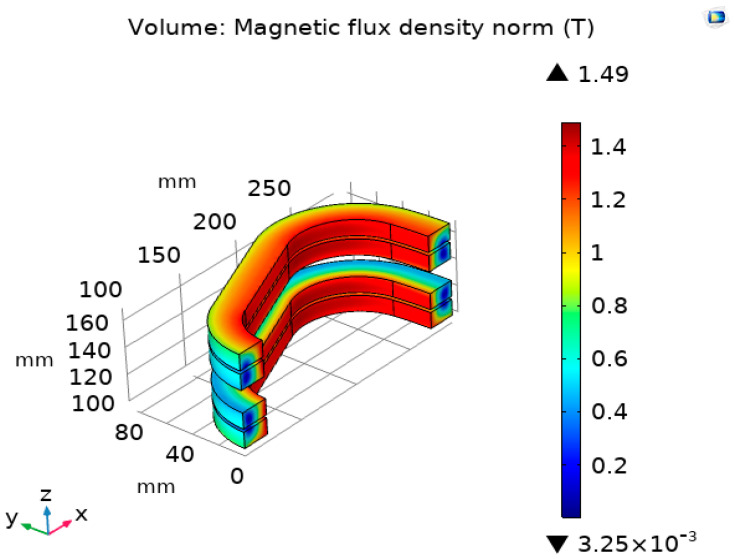
Magnetic field distribution of the HTS field coil.

**Figure 5 materials-16-03587-f005:**
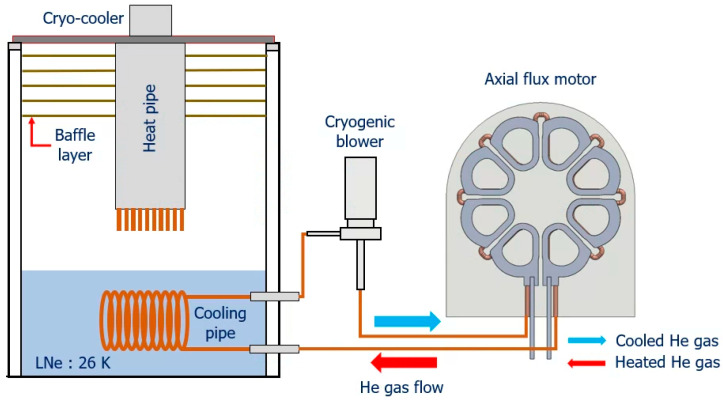
The configuration of the cooling system for the axial flux HTS motor.

**Figure 6 materials-16-03587-f006:**
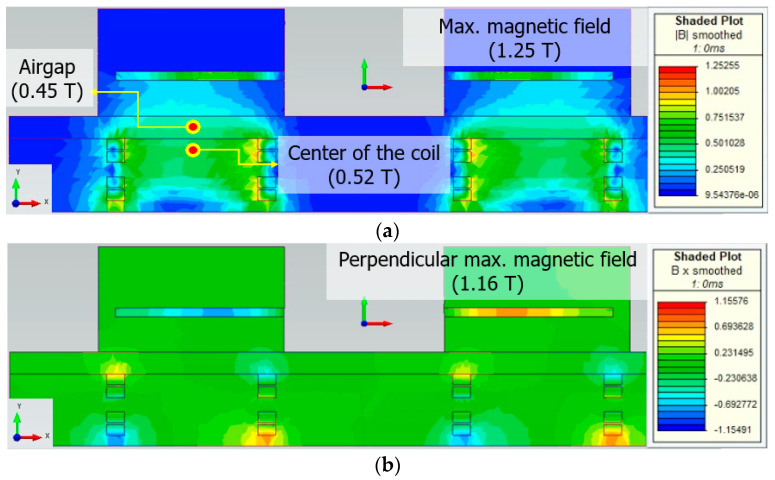
(**a**) Magnetic field distribution and (**b**) perpendicular magnetic field of the axial flux HTS motor.

**Figure 7 materials-16-03587-f007:**
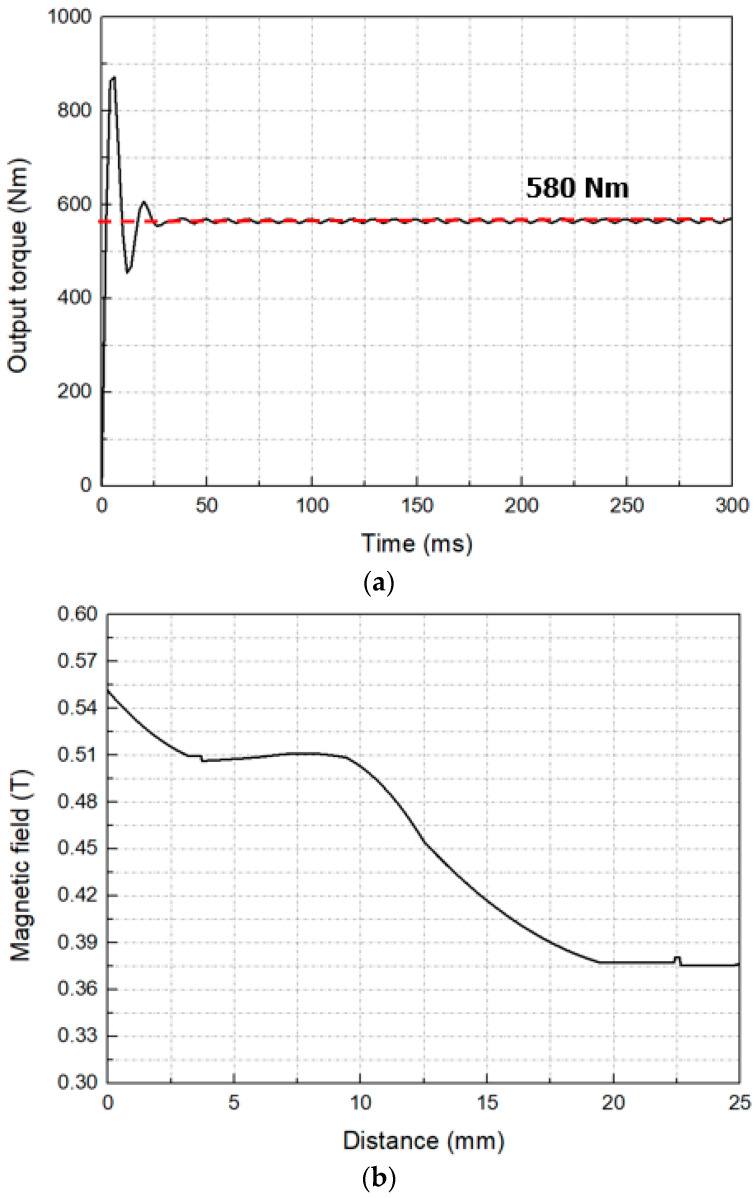
(**a**) Output torque and (**b**) airgap magnetic field of the axial flux HTS motor.

**Figure 8 materials-16-03587-f008:**
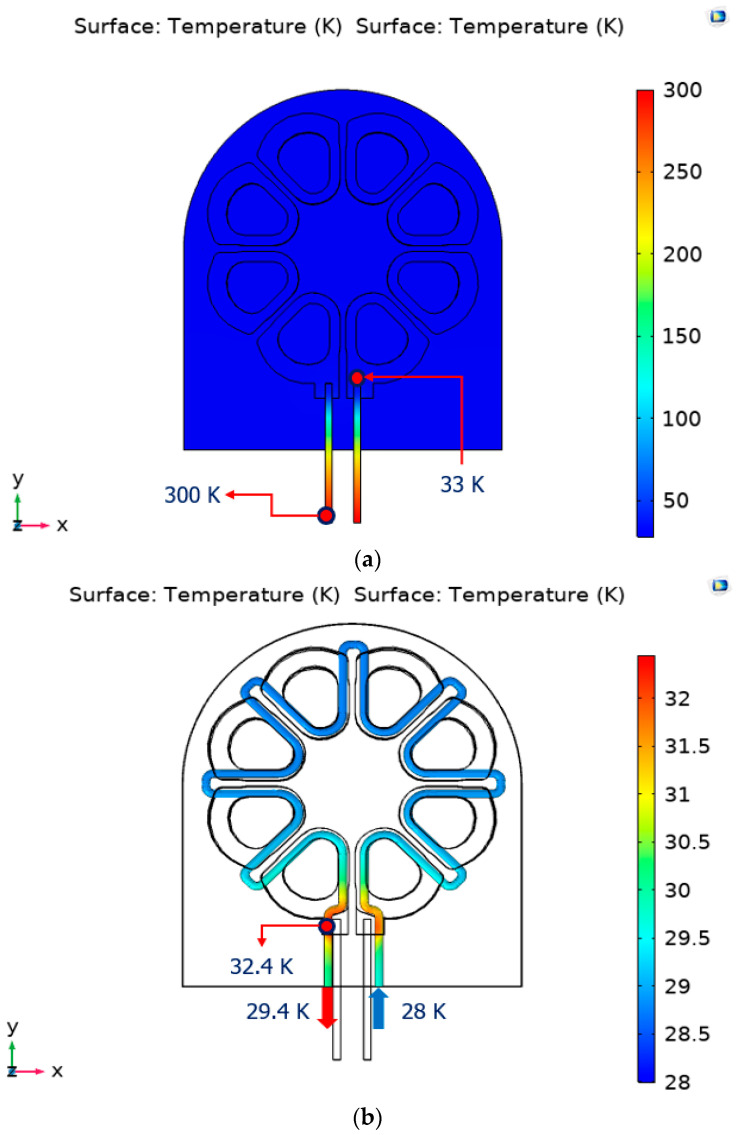
The temperature distribution of (**a**) the cooling system and (**b**) the cooling pipe.

**Figure 9 materials-16-03587-f009:**
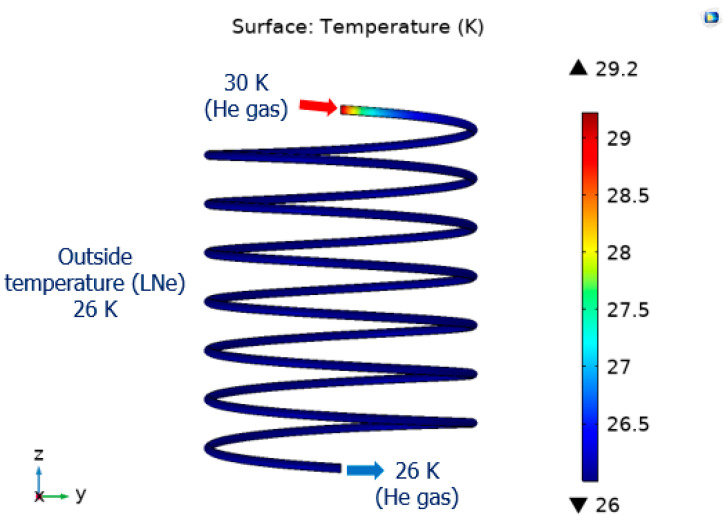
The temperature distribution of heat exchanger.

**Figure 10 materials-16-03587-f010:**
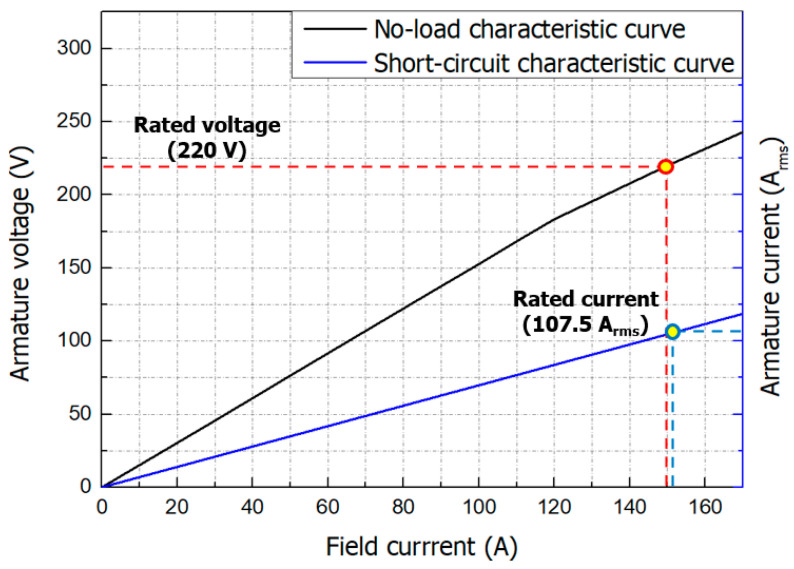
The no-load characteristic curve and short-circuit characteristic curve of the axial flux HTS motor.

**Table 1 materials-16-03587-t001:** Design specifications of the axial flux HTS motor.

Items	Values
Outer diameter of stator and rotor ($D_{o}$)	600 mm
Inner diameter of stator and rotor ($D_{i}$)	180 mm
Mechanical airgap (g)	8 mm
Rated output power	120 kW
Rated L-L voltage	220 V
Rated armature current	107.5 A_rms_
Rated rotational speed	2000 rpm
Rated torque	580 N·m
Operating temperature	35 K
Number of poles	8
Number of field coil layer	4
Number of turns of field coil	130 turns
Number of slot	24
Number of turns of stator coil	54 turns
Total length of the HTS wire	2.09 km

**Table 2 materials-16-03587-t002:** Characteristics of the GdBCO wire.

Items	Values
Width	12 mm
Thickness	0.15 mm
Critical bend diameter	35 mm
Critical tensile stress	500 MPa
Critical tensile strain	0.40%

**Table 3 materials-16-03587-t003:** Specifications of the current lead.

Items	Values
Warm-end temperature ($T_{H}$)	300 K
Cold-end temperature ($T_{L}$)	35 K
The current lead of length (L)	200 mm
Cross section (A)	50 mm^2^
Operating current (I)	150 A

## Data Availability

Data is contained within the article.

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
