# Peer review of "Design and Characteristic Analysis of an Axial Flux High-Temperature Superconducting Motor for Aircraft Propulsion"

_materials, 2023, doi:10.3390/ma16093587_

Round 1

Reviewer 1 Report

The paper reports a design of a 120 kW motor for aircraft. The paper itself is well organized, but I would like the authors to clarify the following points.

1. Please clarify the positioning of the 120 kW capacity. Is the target one of the distributed motors of a large passenger plane or a small airplane? This will lead to different strategies and restrictions on cooling systems.

2. The authors proposes an axial type motor, but please compare it in detail with other reports and claim its superiority. There are many papers on the design of motors for aircraft, so I think this is important to claim originality.

3. While many other reports or omit the cooling system, I think it is great that the authors consider the cooling system. Please include this in your discussion and compare it even with non-superconducting motors.

Reviewer 2 Report

The manuscript deals with the design and characteristic analysis of an axial flux motor with high temperature superconductor (HTS) as field windings, which aims to realize the lightweight and integrated design of aircraft propulsion system. In this paper, a proper review of previous work is presented. And the topology and principle of this mechanism are elaborated in detail. Through the modeling and simulation by FEM, the characteristics are analyzed including magnetic field, temperature field, non-load and short-circuit operation. Overall, the research depth of the manuscript needs to be improved.

Now, the regarding the information presented:

1.       In page 3, Table 1, and Table 2.

For the electromagnetic design of axial flux motor, the dimensions of internal and external diameters are important, as well as air gap. It is suggested to supplement the design parameters in more detail to further reflect the compact and miniaturized design.

2.       In page 5, line 154.

This device is designed and applied to the aircraft propulsion system. The operation conditions should be relatively complex. Why is the variable of TH only considered to be room temperature?

3.       In page 6, line 165 to 168.

In this section, the electromagnetic characteristics of the motor are analyzed. The air-gap magnetic field is 0.45 T. Compared with conventional motor schemes, where is the performance improvement? The comparative analysis should be carried out. Meanwhile, for the electromagnetic analysis, what kind of working condition is analyzed?

4.       In page 6, line 174 to 176.

Based on the electromagnetic analysis results given, the conclusion that “which shows that the axial flux HTS motor can perform stably…” is insufficient. For electromagnetic analysis of motor design, the results of air-gap magnetic field curve, EMF, and force or toque shall also be provided. That is, the axial flux HTS motor is designed for aircraft propulsion, so the thrust and acceleration performance of the motor should be reflected in the electromagnetic analysis. But it is not reflected.

5.       In page 6, Figure 5.

For the analysis of perpendicular magnetic field, it is suggested to illustrate the characteristics by adding the coordinate system in Figure 5.

6.       In page 7, the Section 3.2.

In the thermal analysis of the axial flux HTS motor, only the thermal characteristics of superconducting coils are considered. However, due to the compact multilayer structure, the thermal coupling analysis of the multiphase armature is also important.

7.       In page 8, the Section 3.3.

It is suggested to further explain the research significance of the performance test of the axial flux HTS motor. Meanwhile, according to the no-load characteristic curve of the motor, the overall characteristics of the motor are not further analyzed.

8.       Please put the obtained results in a larger context. For example, the power density achieved should be discussed. That is, compared with the classical machine, what is the efficiency and weight (power density) of the machine proposed?

9.       Overall, the theoretical analysis of the machine is relatively weak. Generally, 3-D finite element modeling and simulation are more suitable for axial flux motor research.

10.   To enhance the practical value, please add more experimental detail.

Reviewer 3 Report

This paper shows the project of an axial flux superconducting electric machine intente for use in aircrafts. The originality of this machine is that that the armature is not superconducting is rotating, and the flux generator are static superconducting coils, in order to facilitate the refrigeration of superconducting coils. But the revolving armature will need sliding rings to work, and this is big problem.

I fact, the paper only presents simulations, without any details. In some cases, the author induces the reader to believe that that the simulations are experimental.

This paper could only be accepted is these issues becomes clear.

Reviewer 4 Report

This manuscript demonstrated a new aircraft propulsion proposal to meet the demand for less pollution and higher efficiency. This propulsion combines the axial flux motors and superconductors. Electric propulsion vehicles are of great help to environmental problems due to their non-polluting and low-noise characteristics, and axial flux high temperature superconducting motor can effectively reduce their volume and weight in the aircraft. The authors explained their experimental purpose, experimental content, experimental method and analyzed the electromagnetic and thermal properties of the motor. However, there are still some issues need to be addressed. Therefore, I think this paper can be considered to be publication after a minor revision.

1.       The author fully explains the charts appearing in the text, but the location of these explanations is always erratic. There should also be a moderate explanation below the chart so that readers can get the information of the chart more quickly. For example, when I just got the article, I was very interested in Figure 5, but the notes below Figure 5 are very brief. If I want to specifically understand the content of Figure 5, I need to read a lot of paragraphs to know that the two paragraphs above Figure 5 are talking about the information of Figure 5.

2.       Some of the expressions in Figure 5 are confused. In the consist of the axial flux HTS motor,whether the "buffer layer" In the article and "baffle layer" in the graph are the same thing, and whether the "cooling pipe" corresponds to the "Heat pipe" in the graph.

3.       Some images are not clear enough in this article, especially in Figure 7. It is suggested to improve the clarity of the images so that readers can have a better reading experience.

4.       What is the basis for the selection of controllable parameters (for instance diameter of stator and rotor, number of turns of coil) in the experiment, and what effect will be caused by changing these parameters on the experimental results.

5.       In the conclusion part, the author sorts out the various performance parameters obtained from the experiment and points out the possibility of applying the axial flux HTS motor to practice. However, the experimental data are not compared with the parameters of the aircraft in use, so it is impossible to judge the feasibility of using axial flux HTS motor as an aircraft propulsion.

Round 2

Reviewer 3 Report

The authors did not followed my suggestions.
